# Preliminary Analysis of Printed Polypropylene Foils and Pigments After Thermal Treatment Using DSC and Ames Tests

**DOI:** 10.3390/ma18143325

**Published:** 2025-07-15

**Authors:** Lukas Prielinger, Eva Ortner, Martin Novak, Lea Markart, Bernhard Rainer

**Affiliations:** Section of Packaging and Resource Management, Department Applied Life Sciences, Hochschule Campus Wien, Favoritenstraße 222, 1100 Vienna, Austria; eva_maria.ortner@hcw.ac.at (E.O.); martin.novak@hcw.ac.at (M.N.); lea.markart@hcw.ac.at (L.M.); bernhard.rainer@hcw.ac.at (B.R.)

**Keywords:** plastics, polypropylene, packaging materials, packaging labels, printed foils, printing inks, pigments, packaging safety, differential scanning calorimetry, miniaturised Ames test, OECD 471, mechanical recycling, food contact materials, EU regulations, EFSA, circular economy

## Abstract

In order to recycle plastic waste back to food contact materials (FCMs), it is necessary to identify hazardous substances in plastic packaging that pose a toxicological risk. Printing inks on plastics are not yet designed to withstand the high heat stress of mechanical recycling processes and therefore require hazard identification. In this study, virgin polypropylene (PP) foils were printed with different types of inks (UV-cured, water-based) and colour shades. Thermal analysis of printed foils and pigments was performed using differential scanning calorimetry (DSC). Samples were then thermally treated below and above measured thermal events at 120 °C, 160 °C, 200 °C or 240 °C for 30 min. Subsequently, migration tests and miniaturised Ames tests were performed. Four out of thirteen printed foils and all three pigments showed positive results for mutagenicity in miniaturised Ames tests after thermal treatment at 240 °C. Additionally, pre-incubation Plate Ames tests (according to OECD 471) were performed on three pigments and one printed foil, yielding two positive results after thermal treatment at 240 °C. These results indicate that certain ink components form hazardous decomposition products when heated up to a temperature of 240 °C. However, further research is needed to gain a better understanding of the chemical processes that occur during high thermal treatment.

## 1. Introduction

Meeting the European Union’s (EU) Green Deal target for the recycling of plastic packaging requires at least 55% of all plastic waste to be recycled by 2030 [1]. In particular, Article 7 of the Packaging and Packaging Waste Regulation (PPWR) sets the minimum percentage of recycled content to 10% for contact sensitive packaging made of plastic materials such as polyolefins [2]. A prerequisite is that recycled plastics intended for use as FCMs must comply with EU Regulation 1935/2004, which states that FCMs must not endanger human health [3]. Consequently, the safety assessment of recycled plastics becomes more important and must be advanced to fulfil current and future safety standards set by the European Food Safety Authority (EFSA) [4]. According to EFSA, the Threshold of Toxicological Concern (TTC) concept applies, which requires unidentified substances in FCMs to be evaluated as potentially direct DNA-reactive. Therefore, a toxicological limit of 0.0025 µg/kg body weight per day is applied [4,5]. In particular, non-intentionally added substances (NIAS) [6] have to be regarded as a safety risk if they are not identified and sufficiently toxicologically evaluated [7,8]. Particularly in post-consumer recycled (PCR) plastics, unidentified NIAS are a risk factor [9] as the required limits of detection (LOD) are very difficult to reach with current analytical methods [10,11,12].

Complementarily, the Ames test is an in vitro bioassay, which is used to specifically screen for direct DNA-reactive effects, achieving very low limits of biological detection (LOBD) for many known mutagenic substances [10,13]. Moreover, the Ames test allows a complex mixture (e.g., FCM extract) to be tested for direct DNA-reactive effects in one run. Mayrhofer et al. [11] reported the number of Ames positives in PCR plastic waste, in particular high-diffusive plastic materials such as polyolefins, increased after mechanical recycling compared to the number of positives before recycling. Even behind functional barriers such as ethylene vinyl alcohol (EVOH), Ames test results remained positive for various mechanically recycled low-density polyethylene (LDPE) films [12]. The only exception was recycled polyethylene terephthalate (PET), which has not yet shown positive results in the Ames test [11,14].

Researchers assume that certain residues in collected plastic waste degrade to potentially hazardous substances after mechanical recycling [9,11]. One of the reported causes are printing inks on plastics, which may form critical degradation products after mechanical recycling processes [11,15,16,17]. Many flexible plastic materials are directly printed or have printed labels attached. These inks and labels lead to an increased number of NIAS if ink residues are not efficiently removed before extrusion [18,19,20]. Lisiecki et al. [16] showed that specific binders and pigments already degrade at around 200 °C, which is below the common extrusion process temperatures for polyolefins of up to 250 °C [21,22]. Moreover, thermogravimetric analysis (TGA) of raw pigments showed significant mass losses between 150 °C and 250 °C, which is caused by the decomposition of monoazo dyes to hazardous compounds such as benzene.

Therefore, the aim of this study is to measure changes in heat capacity (thermal events) induced by phase transitions or degradation of printing inks and pigments with DSC and to measure direct DNA-reactive effects using Ames tests. Two unprinted and thirteen printed PP foils with different types of inks (UV-cured, water-based) and three pigments (blue, red, yellow) were analysed below and above temperatures where DSC measurements yielded characteristic thermal events. This investigation acts as a guide and an indication of critical thermal events where ink components degrade and may form hazardous substances. Based on the results, measures to improve the safety of recycled plastics and alternative packaging for PCR recyclates in less sensitive applications than FCMs are discussed. In addition, conclusions and opportunities for further research to identify critical ink components are provided.

## 2. Materials and Methods

### 2.1. Materials

Virgin polypropylene (PP) foils (PP TOP CLEAR, Spec Code AA661, 50 µm thickness) manufactured by Avery Dennison Materials Group Europe (Oegstgeest, The Netherlands) were supplied by a label manufacturer based in Austria. The material consists of a bi-axially oriented, transparent PP film with a topcoat optimized for excellent ink adhesion, combined with a solvent-based acrylic adhesive (S700) on a transparent 30 µm bi-axially oriented polyester liner (PET30). The foils were printed with eight UV-cured and five water-based inks (with acrylic binders) specifically developed for plastic packaging labels. The selected inks were based on the cyan, magenta, yellow and key (black) (CMYK) colour model [23]. Additionally, three azo pigments (blue (B), red (R) and (golden) yellow (G)), which are commonly used in printing ink formulations, were analysed. Due to confidentiality agreements with the manufacturer, physical images and detailed specifications of the printing inks and pigments cannot be disclosed. Therefore, only anonymised sample codes, including treatment temperatures (in °C), colour shades, and sample types, are provided in Table 1.

### 2.2. Methods

#### 2.2.1. Printing the Foils

The PP foils (PP TOP CLEAR, Spec Code AA661) were printed with acrylic binder systems using laboratory-scale printing methods. Only the PP film was printed. The polyester liner (PET30) was removed prior to the DSC and migration experiments.

For the eight UV-cured inks (C1–C2, M1–M2, Y1–Y2, K1–K2), a film printing system (C1, IGT Testing Systems, Amsterdam, The Netherlands) was used with an application thickness of 0.15 µm. The printed foils were dried using a UV dryer (AKTIPRINT MINI, Technigraf, Grävenwiesbach, Germany) under the following parameters: UV lamp power: 80 W/cm; wavelength: 200 nm; conveyor belt speed: 10 m/min.

For the five water-based inks (M3, Y3–Y4, K3–K4), a film applicator (Coatmaster 510 Basic-V, Erichsen, Hemer, Germany) was used with an application thickness of 0.15 µm. The printed foils were dried overnight at room temperature and stored under dark conditions until further analysis.

#### 2.2.2. Differential Scanning Calorimetry

Thermal analysis of printing inks and pigments was carried out using a DSC 4000 (PerkinElmer Inc., Waltham, MA, USA). Prior to the measurements, water-based printing inks were pre-dried in aluminium crucibles at 50 °C for 2 h to ensure consistent sample conditions with a mass between 5–10 mg [24]. Measurements were carried out in the temperature range of 50 °C to 250 °C, which reflects thermal conditions of a mechanical recycling process of plastics [22]. The lowest possible heating rate of 5 °C per minute was used to simulate precise thermal treatment. All experiments were conducted in a nitrogen atmosphere (20 mL/min) to avoid thermooxidation and to ensure the accuracy of the thermal transition measurements.

Changes in heat capacity (Cp), visible as thermal events, such as phase transitions, cross-linking or thermal decomposition, were recorded and analysed using Pyris™ software (version 13.4; PerkinElmer Inc., Waltham, MA, USA). Characteristic thermal events were identified by evaluating peaks with areas greater than 5 mJ and heights exceeding 0.1 mW. For further analysis, the onset temperature (T_onset_), the temperature of a peak maximum (T_max_) and the offset temperature of a peak (T_offset_) were calculated in °C and listed in Table 2. T_onset_ was defined as the temperature at which a thermal event began, T_max_ was the temperature of a peak maximum and T_offset_ was the temperature at which a thermal event ended. It should be noted that the DSC measurements were carried out under dynamic heating conditions, and therefore the obtained Cp values represent approximate heat capacities only. As such, they allow for qualitative interpretation of thermal behaviour rather than the determination of absolute isothermal heat capacity values. This approach was chosen because the aim of the analysis was not to determine precise Cp values, but rather to identify characteristic thermal transitions that indicate changes in the chemical or physical structure of the material which are further tested with the miniaturised Ames test.

#### 2.2.3. Thermal Treatment

For the printed foils, 300 cm^2^ of foil was used which was cut into 2 × 2 cm^2^ pieces. For the pigments, 250 mg of sample material was used. To simulate thermal treatment similar to a mechanical recycling process, each sample was treated using a Binder heating chamber model FD 115 (Binder GmbH, Tuttlingen, Germany) using the following protocol. First, the samples were heated in an aluminium-lined glass beaker under a nitrogen atmosphere to minimise thermooxidation at a temperature of 240 °C for 30 min. These time-temperature conditions were chosen based on the authors’ experience and the literature analysis [22,25]. The aim was to simulate the high heat stress of a mechanical recycling process and to cover the extrusion temperatures of common thermoplastics, including polyolefins such as PP. Samples that showed a positive result in the miniaturised Ames test were also treated at three additional temperatures, namely 120 °C, 160 °C, or 200 °C for 30 min each. These additional temperature treatments were selected to expose the samples to conditions below and above critical thermal events identified by DSC analysis. The aim was to investigate whether these transitions in the heat capacity of ink components correspond to critical degradation processes that can be detected using the miniaturised Ames test.

#### 2.2.4. Migration Test and Pre-Concentration

Sample migration for printed foils was performed according to EU Regulation 10/2011 [26] and Rainer et al. 2019 [27] with adjustments towards an overestimated migration. Samples were directly exposed to the migration simulant (including the printed surface), which does not simulate migration from printed labels into the packaged product with additional layers. For unprinted and printed PP foils, 300 cm^2^ of foil and 300 mL of 95% ethanol (diluted from ethanol absolute ≥99.8%, AnalaR NORMAPUR^®^ ACS, Reag. Ph. Eur., VWR, Radnor, PA, USA) were filled in Schott bottles with polytetrafluoroethylene (PTFE) caps. Thermally treated PP foils were also migrated in 300 mL of 95% ethanol. Migration was carried out at 60 °C for 10 days. For the pigments, 250 mg of sample and 100 mL of 95% ethanol were filled into Schott bottles with PTFE caps. For comparability, the pigments were also migrated at 60 °C for 10 days.

After migration, the sample volumes were reduced to 1 mL using a Rotavapor^®^ R-300 (BÜCHI Labortechnik AG, Flawil, Switzerland) at 40 °C, 70 mbar and 150 rpm, resulting in an approximate concentration factor of 100 for pigments and 300 for printed foils. A solvent exchange from 95% ethanol to 1 mL dimethyl sulfoxide (DMSO, ≥99%, Merck, Darmstadt, Germany) was performed with a Visiprep TM SPE vacuum manifold DL (Supelco^®^, Bellefonte, PA, USA) at 200 mbar under constant air flow. The sample migrates were then stored at 4 °C until testing.

#### 2.2.5. Miniaturised Ames Test

The concentrated sample migrates were tested using a miniaturised Ames test with *Salmonella* Typhimurium strain TA98 which was supplied by Xenometrix AG (Allschwil, Switzerland). Samples were tested with metabolic activation (+S9) according to the Xenometrix Ames MPF^TM^ manual [28], employing minor adaptations according to a previous protocol from Mayrhofer et al. [11]. All chemicals used for the miniaturised Ames test are listed in Table A1. The bacterial culture was prepared as an overnight culture and was ready for use once the OD_600_ reached between 2 and 6. The OD_600_ was measured using a UV/VIS spectrometer, specifically the Perkin Elmer Lamda XLS (Perkin Elmer, Waltham, MA, USA). The sample migrates were diluted 1:2 with DMSO. The negative control was DMSO and the positive control was 2-AA (2-aminoanthracene, final concentration of 0.5 µg/mL) according to the Xenometrix Ames MPF^TM^ manual [28]. The exposure mix contained 15% S9 mix (consisting of 2.25% S9 and 12.75% cofactors) and 10% bacteria.

The exposure and indicator medium were prepared according to ISO NORM 11350 [29] and the cofactors were prepared according to Hamel et al. 2016 [30]. Detailed percentages of the components used for the exposure-, indicator media and cofactors are provided in Table A2. The S9 fraction (phenobarbital/β-naphthoflavone induced rat liver extract) was supplied by Xenometrix AG (Allschwil, Switzerland). To determine inhibitory effects, each sample was additionally tested with a spike control. The spike was added to the exposure mix in the form of the positive control 2-AA (final concentration of 0.5 µg/mL). After 48 h, the plates were scored according to a colour change from purple to yellow, which indicated a positive well. The miniaturised Ames test is a modification of the standard Plate Ames test according to OECD 471 [31] and differs in several aspects. It is performed in a microtiter plate format and in liquid medium, whereas the standard Plate Ames test uses an agar-based Petri dish approach. In terms of materials and workload, the miniaturised Ames test requires less sample volume (10 µL instead of 50 µL) and offers the possibility of partial automation [32]. Most importantly, the miniaturised Ames test represents an advancement in detecting direct DNA-reactive effects, as it is more sensitive than its agar-based counterpart [13,33].

#### 2.2.6. Pre-Incubation Plate Ames Test

Four selected concentrated migrate samples, which tested positive in the miniaturised Ames test, were additionally tested with a pre-incubation Plate Ames test according to OECD 471 guideline [31] with minor adaptations according to a previous protocol from Rainer et al. 2021 [13]. The Plate Ames test was performed to rule out the possibility that the inherent colour of the tested sample interfered with the results of the miniaturised Ames test, which is based on a colour change in the indicator medium. The preparation of bacterial cultures and the dilution series were performed as described for the miniaturised Ames test. No additional spike control was included in the Plate Ames test. All chemicals used for the Plate Ames test are listed in Table A1. Petri dishes containing glucose minimal agar were prepared and pre-warmed at 37 °C. Top agar was melted, portioned to 2 mL in flip-top tubes and placed in a water bath at 48 °C. Glucose minimal agar and top agar were prepared according to Hamel et al. 2016 [30]. Glucose minimal agar consisted of 1.5% agar agar, 2% Vogel Bonner (VB) salts (50×) and 2.5% glucose solution. Top agar consisted of 0.75% agar agar, 0.75% NaCl and 10% his/biotin solution. Detailed percentages used for the components of the agar types are given in Table A3. DMSO was used as a negative control and 2-AA (final concentration 1.92 µg/mL) as a positive control. 50 µL of the controls and the dilution series were transferred in triplicate to a 24-well plate. In addition, 500 µL of S9 mix (consisting of 55% phosphate buffer, 10.5% cofactors and 1% S9) and 100 µL of bacterial culture were added. As a pre-incubation step, the agar plates were incubated for 90 min at 37 °C and 250 rpm. After exposure, the contents of one well were mixed with 2 mL of top agar, poured and distributed onto a glucose minimal agar plate. The agar plates were then incubated at 37 °C for 48 h before scoring.

#### 2.2.7. Evaluation of the Ames Test Results

Scoring of plates and calculation of n-fold induction for the miniaturised Ames test were performed according to the Xenometrix Ames MPF^TM^ manual [28]. For the pre-incubation Plate Ames test, evaluation was conducted according to the OECD 471 guideline [31] with minor adaptations based on a previous protocol from Rainer et al. 2021 [13]. All numerical evaluations were conducted using Microsoft Excel for Microsoft 365 (Version 2504; Microsoft Corporation, Redmond, WA, USA). Mean values and standard deviations were calculated for each test condition individually to assess reproducibility and variability across replicates. First, the baseline was calculated by adding one standard deviation to the mean number of revertants of the negative control. Then, the n-fold induction was calculated by dividing the mean number of revertants (positive wells or colonies on agar plates) of the sample by the baseline. The n-fold induction was calculated separately for each experiment and if samples exceeded the n-fold induction of 2 (positive threshold), they were classified as mutagenic according to Spiliotopoulos et al. 2024 [34]. To determine the spike recovery, the mean number of revertants of the spiked sample concentration was divided by the mean number of revertants of the spiked negative control. The negative control was defined as 100% spike recovery. If the calculated spike recovery was less than 60%, the sample was classified as inhibiting according to Mayrhofer et al. 2023 [11]. In addition, all results were compared against historical control data and validity thresholds to confirm the expected performance of negative and positive controls [35,36]. This approach also ensured the exclusion of false-positive and false-negative results.

## 3. Results

### 3.1. DSC Results

An overview of the DSC results for selected samples is presented in Table 2. The unprinted PP foil (F1) showed a change in the heat capacity with a maximum at 169 °C, indicating the melting point (T_m_) of the PP foil. In comparison, DSC analysis of a blue UV-cured printed foil (C8) showed a smaller peak at 140 °C (attributed to possible decomposition of the ink) and another peak at 170 °C (T_m_ of the PP foil) (Figure 1).

In Figure 2, two DSC graphs are shown, each depicting the correlation of thermal events with positive Ames test results. A red water-based (WB) ink (M3) showed a thermal event starting at 122 °C (T_onset_), a peak maximum at 123 °C (T_max_) and the end of the thermal event at 150 °C (T_offset_) (Figure 2a). The printed foils with the same ink were thermally treated at temperatures between 120 °C and 240 °C. The miniaturised Ames tests showed a negative result before the thermal event (at 120 °C) and positive results after the thermal event (starting at 160 °C) (Figure 3a,b), indicating that the thermal event led to a critical decomposition process of the ink. A yellow UV-cured ink (Y2) showed a thermal event starting at 201 °C (T_onset_), a peak maximum at 206 °C (T_max_) and the end of the thermal event at 210 °C (T_offset_) (Figure 2b). The miniaturised Ames test showed a negative result before the second thermal event (at 200 °C) and a positive result after the second thermal event (at 240 °C) (Figure 3c,d), also indicating a critical decomposition process of the ink.

Thermal events of other samples did not affect the results of the miniaturised Ames tests. Either no characteristic thermal events were measured, or thermal events showed, for example, melting points of ink components. Some samples (e.g., B1 and G1) showed no thermal events, but the Ames test results were still positive, indicating that not all critical decomposition processes were measured by DSC.

### 3.2. Miniaturised Ames Test Results

#### 3.2.1. Unprinted and Printed PP Foils

An overview of the miniaturised Ames test results is shown in Table 3. Concentrated migrates of two unprinted PP foils (F1 and F2) were analysed: one non-thermally treated (NTT) and one after thermal treatment at 240 °C for 30 min. Both foils showed no inhibitory or mutagenic effects, either untreated or after thermal treatment at 240 °C (Figure 3g,h). Thirteen printed PP foils were tested non-thermally treated and after thermal treatment at temperatures between 120 °C and 240 °C. One printed foil (K4) showed neither inhibitory nor mutagenic effects in the miniaturised Ames test. Eight of thirteen printed foils showed inhibitory effects but no mutagenic effects after thermal treatment at 240 °C. Four printed foils showed mutagenic effects after thermal treatment between 160 °C and 240 °C.

In detail, a magenta water-based printed foil (M3) showed an increased n-fold induction of 5.1 after thermal treatment at 160 °C (Figure 3a,b). Higher thermal treatment increased the n-fold induction to 19.1 at 200 °C and 23.3 at 240 °C. A yellow UV-cured printed foil (Y2) showed only inhibitory effects until 200 °C. After thermal treatment at 240 °C, the sample showed an increased n-fold induction of 3.7 and a positive result for mutagenicity until 25% concentration of the migrate (Figure 3c,d). A yellow water-based printed foil (Y3) also showed only inhibitory effects until 160 °C. After thermal treatment at 200 °C, an increased n-fold induction of 10.7 was measured (Figure 3e,f). Thermal treatment at 240 °C increased the n-fold induction to 16.7. Another yellow water-based printed foil (Y4) showed only inhibitory effects up to 160 °C, but after thermal treatment at 200 °C and 240 °C, the n-fold induction increased to 2.3 and 6.7, respectively.

#### 3.2.2. Pigments

An overview of the miniaturised Ames test results is shown in Table 4. Concentrated migrates of three different pigments (blue (B), red (R), yellow (G)) were tested non-thermally treated (NTT) and at temperatures between 120 °C and 240 °C. The n-fold induction of the blue pigment B1 increased from 3.5 to 7.4 after thermal treatment from 160 °C to 200 °C (Figure 4a,b). Higher thermal treatment increased the n-fold induction to a highest value of 14.1 at 240 °C. The red pigment R1 showed only inhibitory effects at the highest test concentration (at 100%) up to thermal treatment at 200 °C but no mutagenic effects. Higher thermal treatment at 240 °C showed an increased n-fold induction of 2.8 (Figure 4c,d). The yellow pigment G1 showed no mutagenic activity up to thermal treatment at 160 °C. Higher thermal treatments at 200 °C and 240 °C increased the n-fold inductions to 5.6 and 8.4, respectively (Figure 4e,f).

### 3.3. Plate Ames Test Results

Three pigments (B1, R1 and G1) and one printed foil (M3) were tested with the Plate Ames test non-thermally treated (NTT) and after thermal treatment at temperatures between 120 °C and 240 °C (see Table 5). The blue pigment (B1) tested negative for mutagenicity up to thermal treatment at 120 °C. After thermal treatment at 160 °C, the n-fold induction increased to 2.8. The highest increase in n-fold induction of 6.0 was after thermal treatment at 240 °C, indicating a positive result for mutagenicity (Figure 5a,b). A magenta water-based printed foil (M3) showed no mutagenic effects until thermal treatment at 160 °C, but the n-fold induction increased to 3.1 after thermal treatment at 200 °C (Figure 5c,d). The red and yellow pigments (R1 and G1) tested negative when non-thermally treated (NTT) and after thermal treatment at 240 °C.

## 4. Discussion

### 4.1. Positive Ames Test Results After Thermal Treatment of Printed Foils and Pigments

The positive Ames test results in this investigation indicate the formation of hazardous decomposition products of ink components during harsh thermal treatment at temperatures of up to 240 °C. Galbiati et al. [37] listed more than 6000 substances used in printing inks and adhesives applied in plastic food packaging. Some of these substances may not be suitable for high thermal treatment, resulting in the formation of hazardous degradation products. One out of eight UV-cured and three out of five water-based printed foils and all three pigments scored positive results for mutagenicity after thermal treatment at 240 °C in the miniaturised Ames test. Temperatures of up to 250 °C are typically used in the mechanical recycling of polyolefins [22], implying that degradation of some components occurs below the extrusion temperature. This indication is in agreement with a study by Lisiecki et al. [16], revealing that some binders and pigments form critical degradation products at temperatures around 200 °C.

A study by Seoudi et al. [38] found similar results for a phthalocyanine blue pigment. TGA revealed initial decomposition reactions at temperatures below 300 °C, starting at 220 °C. Schreiver et al. [39] reported that the pyrolysis of a phthalocyanine blue pigment led to the formation of benzene, a substance with genotoxic effects [40]. Az et al. [41] found that heat treatment of yellow and red pigments above 200 °C led to the formation of degradation products, such as 3,3′-dichlorobenzidine (DCB) [42]. Similarly, Yan et al. [18] reported that heat treatment of a yellow pigment above 200 °C also leads to the formation of DCB, which is a carcinogenic substance. DCB is a primary aromatic amine (PAA), belonging to a group of chemicals (PAAs) that are considered to be concerning NIAS. PAAs are reported to be one of the degradation products of azo pigments [15,16,18]. In this study, the red (R1) and yellow (G1) pigments tested positive after heat treatment at 200 °C, which could be due to the formation of potentially carcinogenic PAAs. Another possible explanation for positive Ames test results is that nitrocellulose-based inks can degrade to mutagenic nitrosamines when thermally treated in the presence of secondary amines [43].

Further systematic studies with clearly defined compositions, using additional methods such as TGA coupled with gas chromatography–mass spectrometry (GC–MS) or Fourier transform infrared spectroscopy (FTIR) [44,45] are needed to gain more knowledge about critical degradation processes of ink components. If critical ink components that are heat-sensitive are identified using additional analytical methods, it may be possible to improve the toxicological safety of printed PCR plastics by removing or replacing these thermally unstable ink components.

### 4.2. Impacts on the Safety Assessment of Recycled Plastic Materials

The findings of this and other studies [9,11,12,20,46,47] revealed that virgin and recycled plastic materials differ greatly in terms of their hazard potentials. However, due to the huge variety of plastic materials and additives in the packaging industry, the complexity of screening toxicologically relevant substances and differentiating between critical and non-critical substances is still a major issue. Particularly after high thermal processes such as extrusion, the amount of unidentified NIAS increases due to fragmentation of residues from input materials [48]. Additionally, thermo-oxidative effects can significantly contribute to the formation and complexity of NIAS if precautions to exclude oxygen during extrusion or other thermal processes are not taken [49]. Areas of concern in regard to mechanical recycling are plastics that are primarily printed with non-food grade inks and which cannot be easily sorted by near infrared (NIR) spectroscopy [50]. Several of the already mentioned PAAs and nitrosamines are known carcinogens [51,52] and must be identified and toxicologically evaluated at an early stage before polyolefin recyclates can be used in contact-sensitive packaging.

### 4.3. Measures to Improve the Safety of Plastic Recyclates

A prerequisite for achieving a higher quality of PCR plastics is the establishment of a suitable design for mechanical recycling processes based on existing guidelines [53]. This should be applied before taking action to remove unwanted contaminants, as it is currently not feasible to identify all critical NIAS at the proposed EFSA thresholds [4]. If unwanted contaminants from plastic materials are still detected, measures to prevent the formation of hazardous substances are necessary. After melting and granulation of plastics in an extruder, critical substances can no longer be easily removed. Measures such as de-labelling [54] or de-inking [55,56] may improve the safety of PCR plastics by reducing ink components in the final recyclates. Additionally, the use of functional barriers [57] reduces the migration of potentially hazardous substances. An improvement of current sorting technologies is also necessary to sort out primarily printed packaging and to use white or transparent packaging as input material for high-quality recyclates for contact-sensitive packaging [58].

Furthermore, the development of a comprehensive testing strategy to detect all critical substances at the proposed thresholds and to screen batches for suspected hazardous substances and degradation products should be implemented. Additional chemical analysis such as high-performance liquid chromatography coupled with high-resolution mass spectrometry (HPLC–HRMS) is recommended [11,12], and further improvement of the detection limits of currently used in vitro bioassays is required. Improvements in sample preparation and fractionation of extracts using high-performance thin layer chromatography (HPTLC) further reduce the complexity of forests of peaks by separating groups of substances into fractions and thus simplifying their identification by reducing matrix effects [59,60].

### 4.4. Plastic Recyclates for Less Sensitive Applications

If the material quality and safety aspects of mechanically recycled plastics are insufficient for contact-sensitive packaging such as FCM, downcycling to less sensitive applications such as cosmetic packaging is necessary. For cosmetic packaging, there are three main categories, namely leave-on, rinse-off and home care products, with decreasing demands on the quality of plastic recyclates [61]. The recently published DIN SPEC 91521 [62] gives recommendations for the use of recycled plastics with the three levels of suitability as mentioned above and general guidance for analytical methods. Furthermore, a comprehensive guideline for the use of plastic recyclates in cosmetic packaging has been published by the Cosmetics, Packaging and Toxicology (CosPaTox) consortium [61]. For the safe use of a specific percentage of PCR plastic (e.g., 25% recyclate), the maximum acceptable consumer exposure (MACE) for a defined cosmetic packaging must be below the TTC limit of 0.0025 µg/kg bw/day. In addition, the dermal sensitisation threshold (DST) must not be exceeded [63,64], and prohibited substances for cosmetics listed in Annex II of EU Regulation 1223/2009 [65] must be excluded.

A study by Kunita et al. [66] also proposed a risk assessment for the evaluation of cosmetic packaging materials, including the Ames test. If the tolerable migration level (TML) is above the LOBD of the Ames test, the risk of unidentified migrants is negligible in the case of a negative Ames test. However, due to the high number of positive Ames tests in PCR plastics [11], it is still difficult to implement a TML that provides a sufficient margin of safety. Further research is needed to define safe exposure levels for recycled plastics in cosmetic packaging and how to deal with positive Ames test results.

### 4.5. Outlook and Further Research

In order to comply with current EU regulations such as 2022/1616 [67] and PPWR [2], it will be necessary to improve the quality of PCR plastics to enable future applications in FCMs or other contact sensitive applications. The exclusion or removal of thermally unstable ink components should be a priority, as unidentified NIAS pose a risk to consumers’ health. Further research on degradation products of ink components such as binders, pigments and other additives with monitored recycling simulations is also needed. Adhesives and sealants may also pose a risk to mechanical recycling and should be further investigated. Challenge tests for de-inking processes to measure the decontamination efficiencies of surrogates (also called model contaminants [68]) need to be further evaluated, with the aim of extending current standards [55] in regard to safety aspects and expanding analytical test strategies. Aligning current EFSA assessments with those of the Food and Drug Administration (FDA) is also a possible way to approve certain packaging applications for closed-loop recycling [69].

## 5. Conclusions

This study demonstrates that high thermal exposure during mechanical recycling can significantly affect the chemical stability of printing inks and pigments used in plastic packaging, potentially leading to the formation of mutagenic degradation products. The results of the miniaturised Ames test revealed that, after thermal treatment at 240 °C, four out of thirteen printed foils and all three pigment samples showed a positive result for mutagenicity compared to their non-thermally treated counterparts. Additionally, two out of four selected samples (B1 and M3) tested positive in the standard pre-incubation Plate Ames test according to OECD 471 after thermal treatment at 240 °C (Figure 5), confirming the presence of DNA-reactive substances.

Thermal analysis of the printed foils and pigments using DSC revealed characteristic thermal events between 120 °C and 210 °C. However, only two thermal events, associated with inks M3 and Y2, correlated with positive miniaturised Ames test results (Figure 2), indicating that not all thermal events are linked to hazardous degradation processes. These findings underline that ink residues in collected plastic waste may pose a toxicological risk to the safety of PCR plastics if critical ink components degrade during mechanical recycling and cannot be identified.

The results of this investigation also indicate that not all printing inks on packaging labels are resistant to current mechanical recycling processes of plastics, as specific ink components may deteriorate to hazardous degradation products during high-temperature processes such as extrusion. This emphasises the need for ink formulations that can withstand the conditions of the recycling process without producing harmful by-products. Further research is still needed to gain more in-depth knowledge of the chemical processes of ink components that occur during high thermal treatment up to 240 °C.

Furthermore, measures to improve the quality of recycled plastics (such as de-labelling or de-inking) need to be examined in more detail to reduce the formation of critical substances (or generally unidentified NIAS) for the safe use of PCR plastics in FCMs or other contact-sensitive packaging. Identifying and removing critical NIAS in plastic packaging prior to extrusion remains essential to ensure the safety of PCR materials [16,70].

These findings have direct implications for both the packaging and recycling industries, providing practical guidance and highlighting the importance of selecting ink components compatible with recycling conditions. However, the implementation of additional measures to ensure the safety and regulatory compliance of PCR materials used in FCMs and other contact-sensitive packaging needs to be further investigated.

## Figures and Tables

**Figure 1 materials-18-03325-f001:**
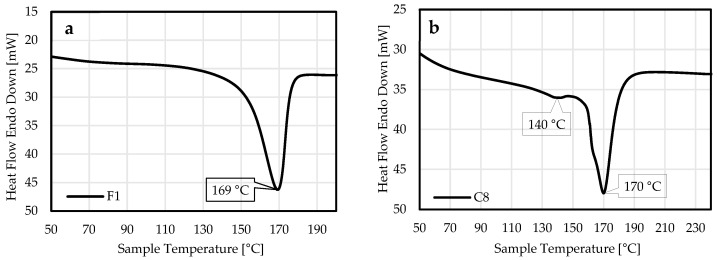
DSC graphs of the unprinted and printed PP foil heated from 50 °C to 250 °C at a heating rate of 5 °C/min. The images show unprinted PP foil F1 (**a**) and blue UV-cured printed foil C8 (**b**) with T_max_ of characteristic peaks.

**Figure 2 materials-18-03325-f002:**
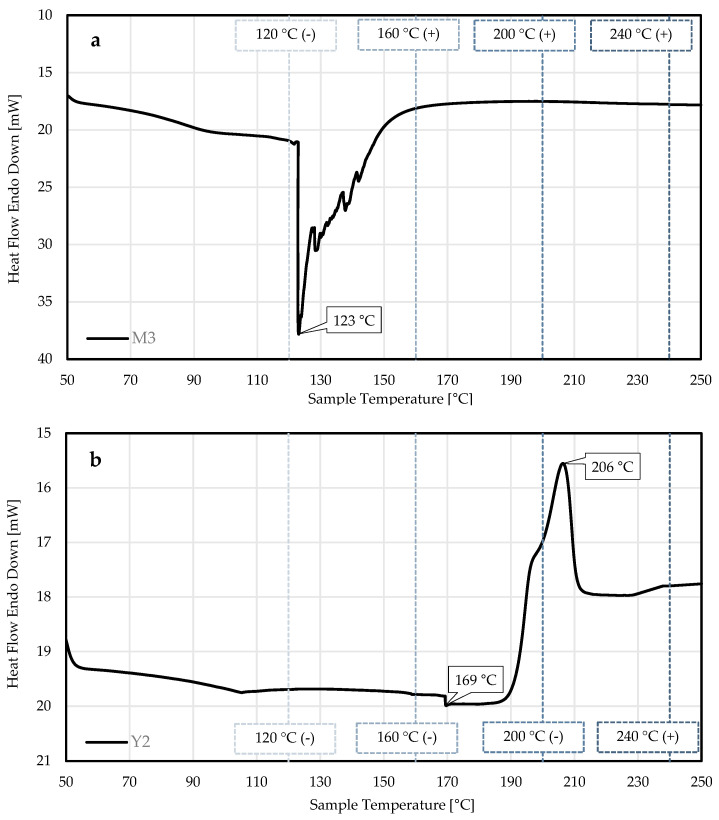
DSC graphs of printing inks heated from 50 °C to 250 °C at a heating rate of 5 °C/min. The plots show (**a**) magenta water-based ink M3 and (**b**) yellow UV-cured ink Y2 with T_max_ of thermal events. The dotted lines show the temperatures between 120 °C and 240 °C applied to the samples prior to the miniaturised Ames tests. A positive (+) or negative (-) Ames test result is indicated by the symbol in the bracket at the respective temperature.

**Figure 3 materials-18-03325-f003:**
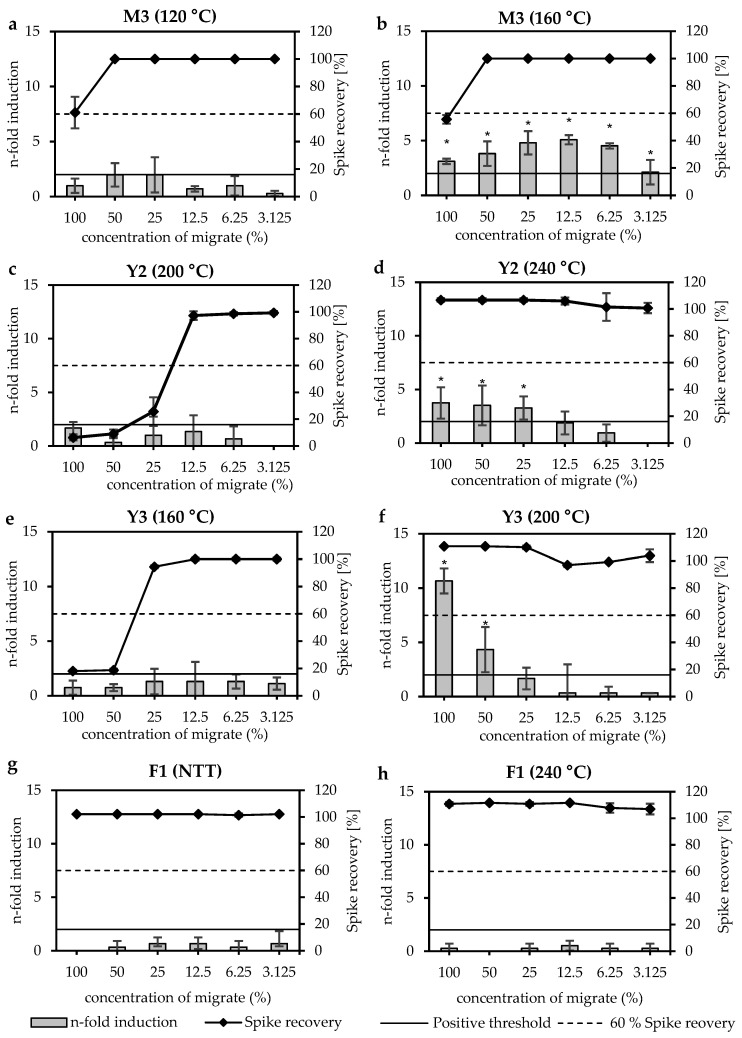
Miniaturised Ames test results of unprinted and printed PP foils with bacterial strain TA98 +S9. The plots show (**a**) magenta water-based printed foil (M3) after thermal treatment at 120 °C, (**b**) the same sample after thermal treatment at 160 °C, (**c**) yellow UV-cured printed foil (Y2) after thermal treatment at 200 °C, (**d**) the same sample after thermal treatment at 240 °C, (**e**) yellow water-based printed foil (Y3) after thermal treatment at 160 °C, (**f**) the same sample after thermal treatment at 200 °C, (**g**) non-thermally treated (NTT) unprinted foil F1, and (**h**) the same sample after thermal treatment at 240 °C. The x-axis shows the concentration of the migrates in %. Bars indicate the mutagenic activity expressed as n-fold induction. Bars marked with a star (*) indicate a positive result. Dots with a continuous line indicate inhibition expressed as spike recovery in %.

**Figure 4 materials-18-03325-f004:**
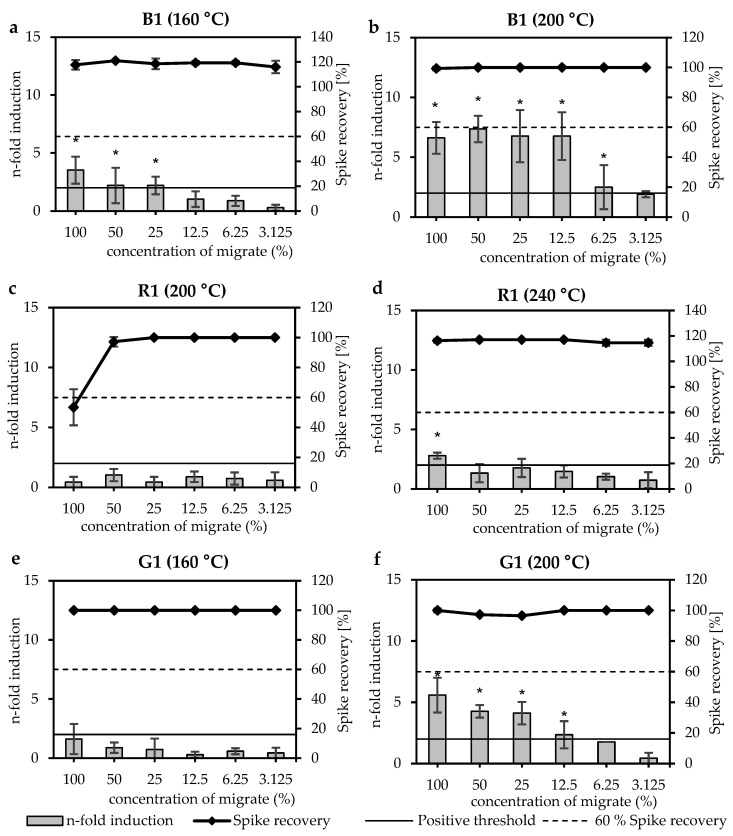
Miniaturised Ames test results of three pigments (blue (B), red (R), yellow (G)) with bacterial strain TA98 +S9. The plots show (**a**) blue pigment B1 after thermal treatment at 160 °C, (**b**) the same sample after thermal treatment at 200 °C, (**c**) red pigment R1 after thermal treatment at 200 °C, (**d**) the same sample after thermal treatment at 240 °C, (**e**) yellow pigment G1 after thermal treatment at 160 °C and (**f**) the same sample after thermal treatment at 200 °C. The x-axis shows the concentration of the migrates in %. Bars indicate the mutagenic activity expressed as n-fold induction. Bars marked with a star (*) indicate a positive result. Dots with a continuous line indicate inhibition expressed as spike recovery in %.

**Figure 5 materials-18-03325-f005:**
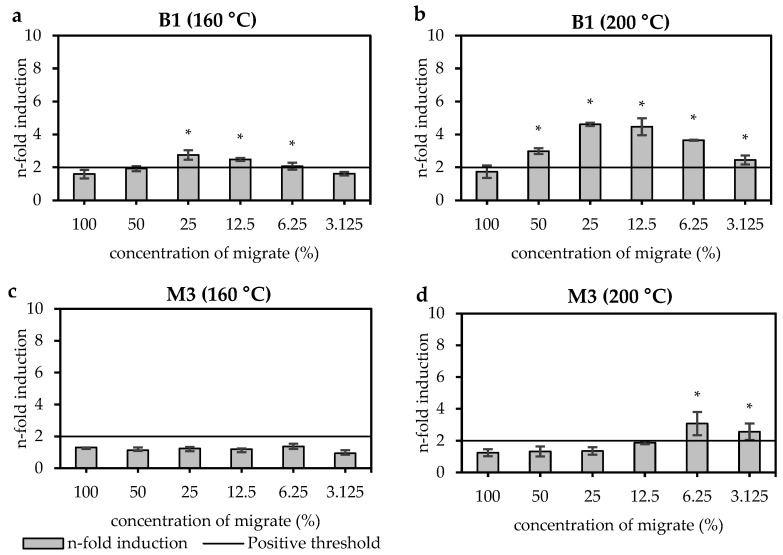
Plate Ames test results of a blue pigment and magenta water-based printed foil with bacterial strain TA98 + S9. The plots show (**a**) blue pigment B1 after thermal treatment at 160 °C, (**b**) same sample after thermal treatment at 200 °C, (**c**) magenta water-based printed foil (M3) after thermal treatment at 160 °C and (**d**) same sample after thermal treatment at 200 °C. The x-axis shows the concentration of the migrates in %. Bars indicate the mutagenic activity expressed as n-fold induction. Bars marked with a star (*) indicate a positive result.

**Table 1 materials-18-03325-t001:** List of two unprinted foils, thirteen printed foils, and three tested pigments. Non-thermally treated (NTT) and thermally treated samples are shown with the treated temperature (in °C), colour shade, and sample type: F: Foil; C: Cyan; M: Magenta; Y: Yellow; K: Key (Black); B: Blue; R: Red; G: Gold (Yellow); UF: unprinted foil; UV: UV-cured ink; WB: water-based ink; PM: pigment.

Sample	Colour	Type	Sample	Colour	Type
F1 (NTT)	Unprinted	UF	Y4 (NTT)	Yellow	WB
F1 (240 °C)	Unprinted	UF	Y4 (120 °C)	Yellow	WB
F2 (NTT)	Unprinted	UF	Y4 (160 °C)	Yellow	WB
F2 (240 °C)	Unprinted	UF	Y4 (200 °C)	Yellow	WB
C1 (NTT)	Cyan	UV	Y4 (240 °C)	Yellow	WB
C1 (240 °C)	Cyan	UV	K1 (NTT)	Black	UV
C2 (NTT)	Cyan	UV	K1 (240 °C)	Black	UV
C2 (240 °C)	Cyan	UV	K2 (NTT)	Black	UV
M1 (NTT)	Magenta	UV	K2 (240 °C)	Black	UV
M1 (240 °C)	Magenta	UV	K3 (NTT)	Black	WB
M2 (NTT)	Magenta	UV	K3 (240 °C)	Black	WB
M2 (240 °C)	Magenta	UV	K4 (NTT)	Black	WB
M3 (NTT)	Magenta	WB	K4 (240 °C)	Black	WB
M3 (120 °C)	Magenta	WB	B1 (NTT)	Blue	PM
M3 (160 °C)	Magenta	WB	B1 (120 °C)	Blue	PM
M3 (200 °C)	Magenta	WB	B1 (160 °C)	Blue	PM
M3 (240 °C)	Magenta	WB	B1 (200 °C)	Blue	PM
Y1 (NTT)	Yellow	UV	B1 (240 °C)	Blue	PM
Y1 (240 °C)	Yellow	UV	R1 (NTT)	Red	PM
Y2 (NTT)	Yellow	UV	R2 (120 °C)	Red	PM
Y2 (120 °C)	Yellow	UV	R3 (160 °C)	Red	PM
Y2 (160 °C)	Yellow	UV	R4 (200 °C)	Red	PM
Y2 (200 °C)	Yellow	UV	R5 (240 °C)	Red	PM
Y2 (240 °C)	Yellow	UV	G1 (NTT)	Yellow	PM
Y3 (NTT)	Yellow	WB	G2 (120 °C)	Yellow	PM
Y3 (120 °C)	Yellow	WB	G3 (160 °C)	Yellow	PM
Y3 (160 °C)	Yellow	WB	G4 (200 °C)	Yellow	PM
Y3 (200 °C)	Yellow	WB	G5 (240 °C)	Yellow	PM
Y3 (240 °C)	Yellow	WB			

**Table 2 materials-18-03325-t002:** Overview of the DSC results for selected samples showing the sample type (printing ink, pigment, printed or unprinted PP foil), the onset temperature (T_onset_), the temperature of a peak maximum (T_max_) and the offset temperature (T_offset_). Samples without characteristic thermal events are indicated with a minus (-).

Sample	Type	T_onset_ (°C)	T_max_ (°C)	T_offset_ (°C)	Sample	Type	T_onset_ (°C)	T_max_ (°C)	T_offset_ (°C)
B1	Pigment	-	-	-	Y3	WB ink	102	121	143
C1	UV ink	115	125	139	Y4	WB ink	145	147	150
		155	175	189	K1	UV ink	122	141	182
R1	Pigment	144	146	153	K3	WB ink	116	121	124
M1	UV ink	122	157	178	K4	WB ink	86	98	110
M3	WB ink	122	123	150			146	147	149
G1	Pigment	-	-	-	F1	PP foil	149	169	177
Y2	UV ink	169	169	171	F1/C1	Printed PP foil	132	140	146
		201	206	210			160	170	179

**Table 3 materials-18-03325-t003:** Overview of miniaturised Ames test results of two unprinted (F1–F2) and thirteen printed (C1–C2, M1–M3, Y1–Y4, K1–K4) PP foils with bacterial strain TA98 with metabolic activation (+S9). Non-thermally treated (NTT) or thermally treated samples (in °C) are indicated next to the sample name in brackets. Results for mutagenicity (M), highest n-fold induction (n-fold), and inhibition (I) are shown. Percentages in brackets indicate the lowest dilution (shown as concentration of migrate (%)) at which mutagenicity or inhibition was measured. +: mutagenic or inhibitory, -: non-mutagenic or non-inhibitory.

Sample	TA98 + S9			Sample	TA98 + S9		
	M	n-Fold	I		M	n-Fold	I
F1 (NTT)	-	-	-	Y2 (160 °C)	-	-	+(25%)
F1 (240 °C)	-	-	-	Y2 (200 °C)	-	-	+(25%)
F2 (NTT)	-	-	-	Y2 (240 °C)	+(25%)	3.7	-
F2 (240 °C)	-	-	-	Y3 (NTT)	-	-	+(100%)
C1 (NTT)	-	-	+(100%)	Y3 (120 °C)	-	-	+(25%)
C1 (240 °C)	-	-	-	Y3 (160 °C)	-	-	+(50%)
C2 (NTT)	-	-	+(25%)	Y3 (200 °C)	+(50%)	10.7	-
C2 (240 °C)	-	-	-	Y3 (240 °C)	+(6.25%)	16.7	-
M1 (NTT)	-	-	+(100%)	Y4 (NTT)	-	-	+(100%)
M1 (240 °C)	-	-	-	Y4 (120 °C)	-	-	+(50%)
M2 (NTT)	-	-	+(100%)	Y4 (160 °C)	-	-	+(100%)
M2 (240 °C)	-	-	-	Y4 (200 °C)	+(50%)	2.3	+(100%)
M3 (NTT)	-	-	-	Y4 (240 °C)	+(6.25%)	6.7	-
M3 (120 °C)	-	-	-	K1 (NTT)	-	-	+(50%)
M3 (160 °C)	+(3.13%)	5.1	+(100%)	K1 (240 °C)	-	-	+(50%)
M3 (200 °C)	+(3.13%)	19.1	-	K2 (NTT)	-	-	+(25%)
M3 (240 °C)	+(6.25%)	23.3	-	K2 (240 °C)	-	-	-
Y1 (NTT)	-	-	+(100%)	K3 (NTT)	-	-	+(50%)
Y1 (240 °C)	-	-	-	K3 (240 °C)	-	-	-
Y2 (NTT)	-	-	+(25%)	K4 (NTT)	-	-	-
Y2 (120 °C)	-	-	+(25%)	K4 (240 °C)	-	-	-

**Table 4 materials-18-03325-t004:** Overview of miniaturised Ames test results of three pigments (blue (B), red (R), yellow (G)) with bacterial strain TA98 with metabolic activation (+S9). Non-thermally treated (NTT) or thermally treated samples (in °C) are indicated next to the sample name in brackets. Results for mutagenicity (M), highest n-fold induction (n-fold) and inhibition (I) are shown. Percentages in brackets indicate the lowest dilution (shown as concentration of migrate (%)) at which mutagenicity or inhibition was measured. +: mutagenic or inhibitory, -: non-mutagenic or non-inhibitory.

Sample	TA98 + S9			Sample	TA98 + S9		
	M	n-Fold	I		M	n-Fold	I
B1 (NTT)	-	-	-	R1 (200 °C)	-	-	+(100%)
B1 (120 °C)	+(50%)	3.1	-	R1 (240 °C)	+(100%)	2.8	-
B1 (160 °C)	+(25%)	3.5	-	G1 (NTT)	-	-	-
B1 (200 °C)	+(6.25%)	7.4	-	G1 (120 °C)	-	-	-
B1 (240 °C)	+(6.25%)	14.1	-	G1 (160 °C)	-	-	-
R1 (NTT)	-	-	+(100%)	G1 (200 °C)	+(12.5%)	5.6	-
R1 (120 °C)	-	-	+(100%)	G1 (240 °C)	+(12.5%)	8.4	-
R1 (160 °C)	-	-	+(100%)				

**Table 5 materials-18-03325-t005:** Overview of Plate Ames test results of three pigments (B1, R1, G1) and a printed foil (magenta water-based ink M3) with bacterial strain TA98 with metabolic activation (+S9). Non-thermally treated (NTT) or thermally treated samples (in °C) are indicated next to the sample name in brackets. Results for mutagenicity (M) and highest n-fold induction (n-fold) are shown. Percentages in brackets indicate the lowest dilution (shown as concentration of migrate (%)) at which mutagenicity was measured. +: mutagenic, -: non-mutagenic.

Sample	TA98 + S9		Sample	TA98 + S9	
	M	n-Fold		M	n-Fold
B1 (NTT)	-	-	M3 (NTT)	-	-
B1 (120 °C)	-	-	M3 (120 °C)	-	-
B1 (160 °C)	+(6.25%)	2.8	M3 (160 °C)	-	-
B1 (200 °C)	+(3.13%)	4.6	M3 (200 °C)	+(6.25%)	3.1
B1 (240 °C)	+(3.13%)	6.0	M3 (240 °C)	+(25%)	2.5
R1 (NTT)	-	-	G1 (NTT)	-	-
R1 (240 °C)	-	-	G1 (240 °C)	-	-

## Data Availability

The original contributions presented in this study are included in the article. Further inquiries can be directed to the corresponding author.

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
