# Peer review of "Preliminary Analysis of Printed Polypropylene Foils and Pigments After Thermal Treatment Using DSC and Ames Tests"

_materials, 2025, doi:10.3390/ma18143325_

Round 1
Reviewer 1 Report
Comments and Suggestions for Authors
In the thermal polymer processing, packaging industry, and during recycling of plastics, the toxicity and mutagenic potential of hazardous products arising from a thermo oxidative degradation can pose a significant risk to consumers. In order to address these incentives, the authors successfully analyzed the results of heat treatment on the initial PP substrate and the pigments, which are used as printing markers on polymer surface, in order to evaluate evidence of mutagenicity in the corresponding products of partial decomposition. In light of environmental challenges, the novelty of the scientific approach and the practical significance of the manuscript are recognized as quite significant.
The findings presented in the manuscript are logically convincing. The textual abstract reflects the general topics and intentions of the authors. The introductory section provides the reader with a good, relatively comprehensive background on the manuscript topics, which immediately acquaint the expert with the main trends of publication. The illustrative content and supplemental data tables, is quite accurate and well-integrated with the text of the manuscript. The terminology is largely understandable for MDPI Journal readers involved in a polymer science area.
However, in order to ensure that the submission meets the provisions for Materials, authors are requested to make the following amendments:
- Despite the fact that the manuscript's text contains a credible reference to "miniaturized Ames test", the authors are invited to expand the readers' knowledge on the differences between miniaturized and regular Ames techniques.
- Why were the temperature and processing time modes chosen for the initial materials? The justification for the selected exposure of the T-t term window can be based on both the author's experience and the literature analysis.
- Without special precautions avoiding oxygen, extrusion, molding, and recycling of polymer waste occur under conditions which initiate thermooxidation. Please involve this term in the appropriate places.
- The authors are kindly requested to significantly revise the part Conclusions. As an innovative presentation, the manuscript is devoted to the disclosure the heat impact on the composites constituents, potentially leading to mutagenicity. This idea should be emphasized in this section first of all. Therefore, it is not advisable to open the section of achievements with an unresolved issue. Instead, the opening remark could be moved to below and combined with the phrase LL 421-424. In addition, for the convenience of readers of the journal, this section could be divided into several paragraphs.
- Fig.3: Improving the quality of this illustration would be beneficial. The values represented by the height of the column, along with their corresponding error bars in fragments C and E, can be more clearly displayed by increasing the scale of the y-axis. In order to preserve the size of the images, we suggest using axis breaking.
In spite of a short series of the minor amendments presented, the topics of presentation fall within the scope of Materials. After the authors' correction, the manuscript should be promoted for the publication.
Reviewer 2 Report
Comments and Suggestions for Authors
Analyzing the heat treatment of printed plastics is a very meaningful topic.
- The information such as the manufacturer and source of the materials needs to be provided.
- If possible, I hope the author can provide physical pictures of the materials.
- hour or hours should be h, please check the manuscript.
- The authors primarily focus on describing the results in the Results and Discussion section, but lack sufficient interpretation of the underlying mechanisms. Many findings are presented without supporting references or comparisons with existing studies.
- The author needs to provide detailed methods and software used for data statistical analysis.
- The impact of this study on industry needs to be mentioned in the conclusion.
Reviewer 3 Report
Comments and Suggestions for Authors
The article titled “Analysis of Printed Plastic Foils and Pigments After Thermal Treatment Using DSC and Ames Tests” is devoted to an exciting topic, as is the analysis of the potential hazards of pigments in plastic foils.
At a glance, this reviewer has identified some preliminary issues that must be addressed before recommending the paper's acceptance.
First, and since the study is far from providing a complete overview of the substances generated during the degradation of the pigments, the title must reveal such circumstance. A proposal for the title would be something like that: “A preliminary analysis of printed……”.
Second, the title gives the false impression that the study concerns a variety of plastic materials, when in fact the authors have only used polypropylene foils. So, please, indicate this clearly in the title.
And third, but the most important, the authors claim to have used polypropylene foils, but without providing evidence of the fundamental nature of the substrates used. Some information must be provided to make the study traceable, a requirement for any scientific communication. In the absence of this information in the article, it cannot be accepted for lack of traceability. The authors must include some evidence that the foil used is PP and not another polymer. For instance, the trademark of the raw material, supplier, or any experimental evidence (FTIR, DSC, etc.) is mandatory. Please note that the only mention of having employed a material is not evidence in the absence of more details.
Other details that need explanation are the following:
- Bold citation: The method assigned to Munhoz et al. is a method that has been a standard for decades. So, the attribution to these authors provides merit to someone who is not the pioneer of the technique by far. Please do not provide a merit to a practitioner. The merit must be originality. Remove the citation.
- In essence, and from a thermodynamic viewpoint, the heat capacity must be obtained under isothermal conditions. The obtention of this Cp under dynamic conditions is a mere and rough approximation with low scientific value, and so the authors must indicate it clearly in the text. I assume that the software can calculate this, but it is a bold value that cannot be accepted in a scientific discussion. This aspect is fundamental in thermodynamics.
- Please provide how the authors have obtained the printed foil.
- In the absence of TGA studies coupled to chromatography or FTIR, the results are very incomplete, as the authors recognize.
In light of the previous concerns and considering the incomplete study and lack of traceability, this reviewer recommends rejecting and resubmitting the article to address the indicated concerns.
Round 2
Reviewer 2 Report
Comments and Suggestions for Authors
None.
Reviewer 3 Report
Comments and Suggestions for Authors
The authors have made a superb revision by thoroughly and adequately addressing all the concerns raised by this reviewer. Below these lines, this reviewer made comments on each one of the seven concerns:
- The authors have now acknowledged that this work is a preliminary study, which is Ok.
- The authors have changed the title by preceding it with they employ PP.
- The lack of information (and thus traceability) about PP has been addressed by including a DSC spectrum probe, which confirms that they are indeed using PP foils.
- The bold citation has now been removed. Brave.
- The authors have now explained the limitations of using Cp obtained from a dynamic experiment suitably.
- This reviewer understands that the authors cannot provide the route of obtaining the samples, as industrial secrets protect them. I guess.
- The response to point 7 is convincing enough.
The article is suitable for publication in Materials.